# Thermodynamic Analysis of Anomalous Shape of Stress–Strain Curves for Shape Memory Alloys

**DOI:** 10.3390/ma15249010

**Published:** 2022-12-16

**Authors:** Dezső L. Beke, Sarah M. Kamel, Lajos Daróczi, László Z. Tóth

**Affiliations:** 1Department of Solid State Physics, University of Debrecen, P.O. Box 400, H-4002 Debrecen, Hungary; 2Physics Department, Faculty of Science, Ain Shams University, Cairo 11566, Egypt

**Keywords:** shape memory alloys, burst-like recovery, stress–strain instability, martensite stabilization

## Abstract

In some shape-memory single crystals the stress–strain (*σ*~*ε*) curves, belonging to stress induced martensitic transformations from austenite to martensite at fixed temperature, instead of being the usual slightly increasing function or horizontal, have an overall negative slope with sudden stress drops in it. We discuss this phenomenon by using a local equilibrium thermodynamic approach and analysing the sign of the second derivative of the difference of the Gibbs free energy. We show that, considering also the possible nucleation and growth of two martensite structural modifications/variants, the stress–strain loops can be unstable. This means that the overall slope of the uploading branch of the stress–strain curve can be negative for smooth transformation if the second martensite, which is more stable with larger transformation strain, is the final product. We also show that local stress-drops on the stress–strain curve can appear if the nucleation of the second martensite is difficult, and the presence of such local stress-drops alone can also result in an overall negative slope of the stress–strain curves. It is illustrated that the increase of the temperature of the thermal recovery during burst-like transition is a measure of the change of the nucleation energy: the more stable martensite has larger nucleation energy.

## 1. Introduction

It is known that in some shape memory single crystals the stress–strain (*σ*~*ε*) curves, belonging to stress induced martensitic transformations at fixed temperature, can show anomalous shape: e.g., the uploading branch of the stress–strain curve, instead being the usual smooth, slightly increasing function, can have an overall negative slope with sudden stress drops on it [1,2,3,4,5,6,7,8,9,10] (Figure 1). This behaviour shows quite a wide variety. It can be strongly anisotropic and can differ considerably along different crystallographic orientations (see e.g., Figure 1 in [2], where the upper branch of the *σ*~*ε* curves, along [100]_A_ as well as [110]_A_ directions in a Ni_49_Fe_18_Ga_27_Co_6_ single crystal, was normal as well as anomalous, respectively). It is often observed that the transformation is not complete [1,7,8,9], but there are also examples when the whole sample has been transformed [4,10]. Furthermore, the strain recovery of the stress induced martensite to austenite during heating can be very fast (burst-like recovery). For instance, it was observed that the DSC peak of this recovery was only about 10^−3^–10^−5^ degree wide (instead of the usual 1–50 K wide transitions at typical 1–10 K/min rates) and was accompanied with jumping of the sample as well as with audible click [1,2,3,4,5,6,7,8,9,10]. In addition, the DSC peak appeared at higher temperature (by 10–60 K higher) than the corresponding peak of the reverse transformation measured after thermally induced cycling. This indicates that the martensite structure formed during stress induced changes is more stable than the thermally induced one. Both the martensite stabilization, manifested in the shift of the DSC peak to higher temperatures during heating, and the anomalous stress–strain curves, are interpreted by the presence/competition of two different martensitic structural modifications [2,3,9]. These can be denoted as M_1_ and M_2_ structures, respectively. For instance, in Ni_49_Fe_18_Ga_27_Co_6_ single crystal M_1_ and M_2_ were identified as twinned 14 M martensite, as well as L1_o_ detwinned tetragonal martensite, respectively [3], or in Cu–Al–Ni alloys as *β*′ (18 R monoclinic) and/or twinned *γ*′ (2 H orthorhombic) as well as detwinned *γ*′ phases, respectively [7,8,11]. It can be added that the appearance of stress drops/jumps or the macroscopic jump of the sample itself can cause difficulties in experiments. For instance, in ref. [2] it was observed that the DSC peak of the thermal recovery showed an anomalous shift to lower temperatures with increasing heating rates. In [7] it was mentioned that the slow motion of the crosshead of the machine did not go down so fast and a stress drop could be registered or that the testing machine can be slightly deformed itself. However, in these papers the above effects were carefully handled, and it can be concluded that the observations summarized above on the anomalous stress–strain curves and on the burst-like recovery are real effects.

In fact, the above martensite stabilization is similar to the effect of the so-called SIM aging, where the stabilization is achieved by aging under uniaxial stress after the formation of stress induced martensite [11,12,13,14,15,16,17,18]. While the SIM aging effect is well interpreted by the symmetry confirming short range ordering [14,19,20], the interpretation of the anomalous shape of the stress–strain curves and the burst-like strain recovery still is in infancy [1,3,4,9] and mostly qualitative arguments were offered. There are only few articles in which more quantitative treatment can be found from two groups [1,3,21,22,23]. In [1,3,21] the so-called theory of diffuse martensitic transformation is used, which involves a second order phase transformation. The models used in [22,23] are based on first order transformation and explain the anomalous *σ*~*ε* curves only for a transformation between the austenite and one martensite modification. It was obtained in [22] that the slope of the upper branch of the *σ*~*ε* curve can be negative only if $S_{A}-S_{M}>0$, where *S_A_* and *S_M_* are the stiffness of the austenite and martensite, respectively. However, this conclusion contradicts to a set of experimental results where anomalous stress–strain curves were observed for $S_{A}-S_{M}<0$ [3,7,9]. 

Thus, the main subject of our paper is to provide a simple thermodynamic analysis, based on our local equilibrium formalism [24,25]. In contrast to [21], we investigate the stability conditions always for a first order transformation, which is in line with [22,23]. It will be shown that the uploading stress–strain curve always has positive slope if only one martensite forms and grows smoothly (i.e., the martensite volume fraction approximately continuously increases). Negative slope can be obtained for smooth transformations if there is a growth/competition of two martensite structural modifications/variants and if the second (more stable) one is the final product. Furthermore, local stress drops on the *σ*~*ε* curve can appear if the nucleation of the second martensite is difficult. We hope that our results provide an important, new contribution to the understanding of the above frequently observed experimental behaviour, which can also be manifested in burst-like temperature induced strain recovery of the stabilized martensite structure. Such martensite stabilizations are very useful in producing two-way shape memory behaviour and rubber-like behaviour with extended recoverable strains, important in many applications.

## 2. Model Calculations

### 2.1. The Local Equilibrium Formalism

Let us start from our local equilibrium formalism summarized in [25] and used in [24]. In local equilibrium the derivative of the *total* change of the Gibbs free energy per unit volume, ΔG, versus the transformed martensite volume fraction, *ξ*, should be equal to zero. For the forward, (from austenite to martensite) transformation,
(1)$$\frac{\partial \left(\Delta G\right)}{\partial \xi }=\frac{\partial \left(\Delta G_{c}+E+D\right)}{\partial \xi }=\frac{\partial \left(\Delta G_{c}\right)}{\partial \xi }+e\left(\xi \right)+d\left(\xi \right)=0.$$

Here, $\xi =\frac{V_{M}}{V_{M}+V_{A}}$ with $V=V_{A}+V_{M}$ (*V* is the volume of the sample), *E* as well as *D* denote the elastic and dissipative energies per unit volume, belonging to the *A*→*M* (*AM*) transformation and $e\left(\xi \right)=\frac{dE\left(\xi \right)}{d\xi }$ as well as $d\left(\xi \right)=\frac{dD\left(\xi \right)}{d\xi }$, respectively. The total dissipative energy, $D_{t}=\int_{0}^{1}d\left(\xi \right)d\xi ,$ is obviously positive in both directions ($D_{t}^{AM}=D_{t}≅D_{t}^{MA}>0$) and, if the thermoelastic balance is also assumed, then $E_{t}^{AM}=E_{t}=\int_{0}^{1}e\left(\xi \right)d\xi =-E_{t}^{MA}>0.$ Usually, there is one additional term in the expression of Δ*G*, which is the nucleation energy related to the formation of new interfaces during the transformation. Since this is always positive in both directions, like the dissipative energy, it can be considered as it would be included in the dissipative term [24,25].

In Equation (1) ∆*G_c_* is the change in the chemical free energy per unit volume and its derivative can be written as
(2)$$\frac{\partial \Delta G_{c}}{\partial \xi }=\frac{\partial \left[\xi G_{M}+\left(1-\xi \right)G_{A}-G_{A}\right]}{\partial \xi }=\frac{\partial \xi \left(G_{M}-G_{A}\right)}{\partial \xi }=\left(G_{M}-G_{A}\right)+\xi \frac{\partial \left(G_{M}-G_{A}\right)\;}{\partial \xi }$$
where
(3)$$(G_{M}-G_{A})=∆u-∆s-\varepsilon^{\mathrm{tr}}$$

Here ∆*s = s_M_* − *s_A_* (∆*s* < 0) as well as Δ*u = u_M_* − *u_A_ =* Δ*u* (<0) are the entropy and internal energy changes per unit volume, respectively, and they are independent of *ξ*. T and *σ* are the temperature and stress as well as *ε^tr^* is the transformation strain, which is positive for tension.

In the literature it is frequently assumed that the transformation strain, *ε^tr^*, is constant and independent of *ξ* (i.e., the derivative of Equation (3) is zero). However, in general *ε^tr^* can depend on the *T* and/or *σ* values since in Equation (3) the stress-term has a tensor character [24,25]. Thus, even for the application of uniaxial stress (which leads to a scalar term, as in Equation (3)), *ε^tr^* in principle can depend on *T* and *σ* too. In case of formation and growth of two martensite structural modifications, it can also depend on the volume fraction of one of these, *η*(*ξ*) *= η*(*T*,*σ*) *= V_M_*_2_*/V_M_* (*V_M_ = V_M_*_1_
*+ V_M_*_2_, *V = V_M_ + V_A_*), [24,25,26]. The *η* dependence of *ε^tr^* can be given by the relation [24,25,26]:*ε**^tr^*(*η*) = *ε*_1_ + (*ε*_2_ − *ε*_1_)*η*(4)
where *ε*_1_ and *ε*_2_ are the transformation strains when fully one certain martensite structure forms (at *η* ≅ 0 and *η* ≅ 1, respectively). Of course, the details can be very complex and the value of *ε* can also be different for twinned or detwinned martensite variants [27,28,29]. As a consequence, *η can have a ξ-dependence*, *η*(*ξ*), *which explains the ξ-dependence of ε^tr^: ε^tr^*(*η*(*ξ*)). Furthermore, *ε_i_*, due to the different temperature dependence of the elastic moduli of the austenite and martensite phases, can have a direct temperature dependence too [22,30,31]). Thus, considerations on the *ξ*-dependence of the actual strain, $\varepsilon \left(\xi \right)$ (which should be distinguished from the transformation strain, $\varepsilon^{tr}$ can be important for estimation of the *σ*(*ε*) curve (see also below). It is worth mentioning that our description, in order to concentrate on the interpretation of the main important features, in its form is a simplified one, and is applicable indeed for single crystals (with two martensite modifications), where the effects summarized in the introduction were observed. Of course, for more complex systems (e.g., for polycrystalline materials with numerous martensite variants) one would need a more sophisticated treatment, such as the phase field method in the form of the well-known Ginzburg–Landau theory (see e.g., [32]).

### 2.2. Expressions for the $\sigma \left(\xi \right)$
$\left(\sigma \left(\varepsilon \right)\right)$ Functions

In contrast to [24,25], we will assume that *ε^tr^* depends on *ξ*, and we have from (2) and (3)
(5)$$\frac{\partial \Delta G_{c}}{\partial \xi }=\Delta u-T\Delta s-\sigma \varepsilon^{tr}-\xi \sigma \frac{\partial \varepsilon^{tr}}{\partial \xi }=\Delta u-T\Delta s-\sigma \varepsilon^{tr}\left(1+\xi \frac{\partial \varepsilon^{tr}}{\varepsilon^{tr}\partial \xi }\right)$$

Since a similar relation holds for $\frac{\partial \Delta G_{c}^{MA}}{\partial \left(1-\xi \right)}$ (1−ξ is the austenite volume fraction) belonging to the reverse (martensite to austenite, *MA*) transformation [24,25]. In the following we give detailed considerations only for the forward transformation, and expressions with upper indexes *^MA^* as well as *^AM^* will be used only if a comparison of the forward and reverse transformations is made.

Taking Equation (5) equal to zero for *T* = *const.*
(6)$$\sigma_{0}\left(T,\xi \right)=\frac{1}{\varepsilon^{tr}\left(T\right)\left(1+\xi \frac{\partial \varepsilon^{tr}}{\varepsilon^{tr}\partial \xi }\right)}\left[\Delta u-T\Delta s\right]=\sigma_{o}\left(0,\xi \right)-\frac{T\Delta s}{\varepsilon^{tr}\left(T\right)\left(1+\xi \frac{\partial \varepsilon^{tr}}{\varepsilon^{tr}\partial \xi }\right)}$$

Here $\sigma_{o}\left(0,\xi \right)=\frac{\Delta u}{\varepsilon^{tr}\left(T=0\right)\left(1+\xi \frac{\partial \varepsilon^{tr}}{\varepsilon^{tr}\partial \xi }\right)}\;(\Delta u<0,\Delta s\;\leq \;0$) is the equilibrium transformation stress (at T=0). If the transformation strain is independent of *ξ*, the usual form of the well-known Clausius–Clapeyron equation is obtained [24,25], i.e., $\sigma_{o}\left(T\right)=\sigma_{o}\left(0\right)-\frac{\Delta s}{\varepsilon^{tr}}T$.

Furthermore, from the condition (1), using Equations (5) and (6) as well, we obtain the expression for the forward branch of the σ(*ξ*) function (at a fixed value of *T*) as
(7)$$\sigma \left(T,\xi \right)=\sigma_{o}\left(T,\xi \right)+\frac{e\left(\xi \right)+d\left(\xi \right)}{\varepsilon^{tr}\left(T\right)\left(1+\xi \frac{\partial \varepsilon^{tr}}{\varepsilon^{tr}\partial \xi }\right)}$$
(see also [24,25] for constant *ε^tr^*).

The start and finish stresses, for both the forward and reverse transformations, can be given as
$$\sigma_{Ms}\left(T,\xi =0\right)=\sigma_{o}\left(T,0\right)+\frac{e_{o}+d_{o}}{\varepsilon^{tr}}$$
$$\sigma_{Mf}\left(T,\xi =1\right)=\sigma_{o}\left(T,1\right)+\frac{e_{1}+d_{1}}{\varepsilon^{tr}\left(1+\frac{1}{\varepsilon^{tr}}{\left.\frac{\partial \varepsilon^{tr}}{\partial \xi }\right|}_{\xi =1}\right)}$$
$$\sigma_{As}\left(T,\;1-\xi =0\right)=\sigma_{o}\left(T,1-\xi =0\right)+\frac{e_{1}^{MA}+d_{1}^{MA}}{\varepsilon^{trMA}}$$
(8)$$\sigma_{Af}\left(T,\;1-\xi =1\right)=\sigma_{o}\left(T,1-\xi =1\right)+\frac{e_{o}^{MA}+d_{o}^{MA}}{\varepsilon^{trMA\;}\left(1+\frac{1}{\varepsilon^{trMA}}{\left.\frac{\partial \varepsilon^{trMA}}{\varepsilon^{trMA}\partial \left(1-\xi \right)}\right|}_{\left(1-\xi \right)=1}\right)}$$

## 3. Discussion

### 3.1. Expressions for the Widths of Transformations—Investigation of the Stability during Phase Transformation

Let us consider the sign of the second derivative of the difference of the Gibbs free energy: this gives information about the stability of the system against fluctuations in the volume fraction during growth. If it is positive the two-phase system is stable during the transformation. From Equations (1), (4) and (5) we have, for the austenite to martensite, AM, transformation
(9)$${\frac{\partial^{2}\Delta G}{\partial \xi^{2}}\rfloor }_{T}=\frac{\partial^{2}\left(\Delta G_{c}\right)}{\partial \xi^{2}}+\left(\frac{\partial e}{\partial \xi }+\frac{\partial d}{\partial \xi }\right)=-\sigma \left(2\frac{\partial \varepsilon^{tr}}{\partial \xi }+\xi \frac{\partial^{2}\varepsilon^{tr}}{\partial \xi^{2}}\right)+\left(\frac{\partial e}{\partial \xi }+\frac{\partial d}{\partial \xi }\right).$$

It can be seen that for a detailed stability analysis we have to discuss the meaning of the terms on the right had side of Equation (9). 

#### 3.1.1. Meaning of the Elastic and Dissipative Terms

During thermoelastic *AM* transformations the elastic energy, *E*(*ξ*) > 0, is due to the local strain fields around the martensite nuclei formed and due to the overlap of the elastic fields: the latter contribution can be proportional to $\xi^{2}$ [24,25]. Thus, one can assume that
(10)$$\frac{\partial E\left(\xi \right)}{\partial \xi }=e\left(\xi \right)=e_{o}+\left(e_{1}-e_{o}\right)\xi >0$$
and the elastic contribution to Equation (9) is
(11)$$\frac{\partial e}{\partial \xi }=\left(e_{1}-e_{o}\right)\;\geq \;0$$

For the reverse transformation the stored elastic energy is released, i.e., $e^{MA}\left(\xi \right)=-e^{MA}\left(1-\xi \right),$ and we can similarly write
(12)$$e^{MA}\left(1-\xi \right)=e_{o}^{MA}+\left(e_{1}^{MA}-e_{o}^{MA}\right)\left(1-\xi \right)<0$$
and
$$\frac{\partial e^{MA}}{\partial \left(1-\xi \right)}=\left(e_{1}^{MA}-e_{o}^{MA}\right)=-\left(e_{1}-e_{o}\right)\;\leq \;0$$

The dissipative energy, *D*(*ξ*), is also positive for the forward transformation and can be considered as the sum of two terms, $D=D_{f}+D_{n}$. *D_f_* (>0) originates from the frictional-type motion of the interfaces and can be supposed that it is proportional to *ξ*, while $D_{n}$ (>0) is due to the nucleation energy. In the simple case when a large number of martensite nuclei form (smooth transformation [22]) $\;D_{n}$ can also be approximately a monotonic linear function of *ξ* and thus
(13)$$d\left(\xi \right)=d_{o}=d_{1}=const.\;>0$$

Refs. [24,25] and its contributions to (9) is zero. In a more general case, e.g., if the nucleation is difficult and happens suddenly at certain temperatures, $\frac{\partial d}{\partial \xi }=\frac{\partial^{2}\left(D\right)}{\partial \xi^{2}}\;\neq \;0$, and *D_n_*(*ξ*) can be a complicated (step-wise) function of *ξ* (see also below). 

#### 3.1.2. Stress–Strain Loops

First consider smooth thermoelastic transformations, when only one type of martensite structural modification grows, i.e., $\varepsilon^{tr}$ is independent of *ξ* ($\varepsilon^{tr}=const.$) and $\Delta s\;\left(<0\right)=-\Delta s^{MA}$ [24,25]. Then, one gets from Equation (9) with Equation (7) that
(14)$${\frac{\partial^{2}\Delta G}{\partial \xi^{2}}\rfloor }_{T}=\frac{\partial \sigma }{\partial \xi }\varepsilon^{tr}=\left(\frac{\partial e}{\partial \xi }+\frac{\partial d}{\partial \xi }\right)\;≅\;\left(e_{1}-e_{o}\right)=(\sigma_{Mf}-\sigma_{Ms})\varepsilon^{tr}$$
where Equations (11) and (13) were also used. Note, that in this case the second derivative of Δ*G* is proportional to the slope of the *σ*(*ξ*) function and $\left(\sigma_{Mf}-\sigma_{Ms}\right)$ is the width of the upper branch of the schematic hysteresis loop shown in Figure 2. Since, in accordance with Equation (11) $e_{1}-e_{o}\;\geq \;0$, it is usually positive, or close to zero (for horizontal branches), the stability condition fulfils for this simple case. Similarly, we can deduce for the down branch (using also that $\varepsilon^{tr}=-\varepsilon^{trMA}$ and Equation (12))
(15)$${\frac{\partial^{2}\Delta G^{MA}}{\partial {\left(1\;-\;\xi \right)}^{2}}\rfloor }_{T}=\frac{\partial \sigma^{MA}}{\partial \left(1\;-\;\xi \right)}\varepsilon^{trMA}\;≅\;\frac{\partial e^{MA}}{\partial \left(1\;-\;\xi \right)}=-\frac{\partial e^{MA}}{\partial \xi }=\left(e_{1}\;-\;e_{o}\right)\;\;≅\;\left(\sigma_{As}\;-\;\sigma_{Af}\right)\varepsilon^{tr}$$
for which the stability condition also fulfils, since $\left(\sigma_{As}\;-\;\sigma_{Af}\right)\;\geq \;0$ (Figure 2). 

#### 3.1.3. Thermal Hysteresis Loops

Under similar conditions as (14) was obtained we can write for the cooling as well as for the heating branches of the thermal hysteresis loop (i.e., for the $\xi \left(T\right)$ function) [24] that
(16)$${\frac{\partial^{2}\Delta G}{\partial \xi^{2}}\rfloor }_{\sigma =0}=\frac{\partial^{2}\left(E_{T}+D_{T}\right)}{\partial \xi^{2}}\;≅\;\left(\frac{\partial e_{T}}{\partial \xi }\right)\;≅\;\;\left(e_{1}\;-\;e_{o}\right)\;≅\;\left(M_{s}\;-\;M_{f}\right)\left(-\Delta S\right)$$
as well as
(17)$${\frac{\partial^{2}\Delta G^{MA}}{\partial {\left(1\;-\;\xi \right)}^{2}}\rfloor }_{\sigma =0}≅\frac{\partial e_{T}^{MA}}{\partial \left(1\;-\;\xi \right)}≅\left(e_{1T}^{MA}\;-\;e_{oT}^{MA}\right)=-\left(e_{1T}\;-\;e_{oT}\right)=\left(A_{f}\;-\;A_{S}\right)\left(\;-\;\Delta s\right)>0$$

The lower indexes *T* indicate that the derivatives of the elastic (and dissipative) energies can be different for the *ξ*(*T*) and *σ*(*ξ*) functions (see also below). It is clear, that since both $\left(M_{s}\;-\;M_{f}\right)$ and $\left(A_{f}-A_{S}\right)$ are positive the thermal hysteresis loops are always stable. It is worth noting that Equations (16) and (17) are also valid if $\varepsilon^{tr}$ is not constant, since at σ=0 the term $\sigma \left(2\frac{\partial \varepsilon^{tr}}{\partial \xi }+\xi \frac{\partial^{2}\varepsilon^{tr}}{\partial \xi^{2}}\right)$, which is present in Equation (9), cancels out. 

### 3.2. Growth of Two Martensite Modifications

Let us now consider the more general case when $\varepsilon^{tr}$ has *ξ*-dependence as given by Equation (4) with *η*(*ξ*). It is clear from Equation (9) that, if the first term is negative and its magnitude is larger than the last two terms, the system can be unstable. Using Equation (4) we can write
(18)$$\frac{\partial \varepsilon^{tr}}{\partial \xi }=\left(\varepsilon_{2}\;-\;\varepsilon_{1}\right)\frac{\partial \eta }{\partial \xi }$$
and the first term of Equation (9) will be
(19)$$-\sigma \left(\varepsilon_{2}\;-\;\varepsilon_{1}\right)\left[2\frac{\partial \eta }{\partial \xi }+\frac{\partial^{2}\eta }{\partial \xi^{2}}\right]$$

This term can be negative if $\varepsilon_{2}>\varepsilon_{1}$ and the second martensite grows monotonically with *ξ* (smooth transition). Of course, in general, the situation can be more complex. For example, during nucleation and growth of two martensite modifications, when the first one of them grows and suddenly the second one starts to nucleate and grow, there can be sudden changes in the stored elastic and nucleation energies (the nucleation energy is included in D). Thus, in the second derivatives of E and D there can also appear sudden sign changes (jumps), which can be quite local at around a certain value of *ξ*. Another complication is involved in the fact that, in this general case, the martensite volume fraction is not proportional to the actual strain.

Nevertheless, the slope ${\frac{\partial^{2}\Delta G}{\partial \xi^{2}}|}_{T}$ (see Equation (9)) can be negative if the magnitude of the term given by Equation (19) is larger than the value $\left(\frac{\partial e}{\partial \xi }+\frac{\partial d}{\partial \xi }\right)≅\left(e_{1}\;-\;e_{o}\right)$, and then the overall slope of the uploading branch of the *σ*(*ε*) curve can be negative, i.e., the two-phase system can be unstable during the transition. For burst-like recovery, when $A_{s}\;-\;A_{f}\;≅\;0$, $\left(\frac{\partial e}{\partial \xi }+\frac{\partial d}{\partial \xi }\right)\;≅\;\left(e_{1}\;-\;e_{o}\right)$ is close to zero, and in this case the condition of instability can be given as $-\sigma \left(\varepsilon_{2}-\varepsilon_{1}\right)\left[2\frac{\partial \eta }{\partial \xi }+\frac{\partial^{2}\eta }{\partial \xi^{2}}\right]<0.$

### 3.3. Nucleation Difficulties

Let us consider the following simple case: assume that with increasing stress the M_1_ martensite modification nucleates first and grows and at a certain transformed volume fraction, *ξ_c_* (~*ε_c_*) the second martensite M_2_ nucleates. Suppose that M_2_ is more stable than M_1_, in accordance with the higher temperature of burst-like MA transformation and, as above, $\varepsilon_{2}=\varepsilon^{tr2}>\varepsilon_{1}=\varepsilon^{tr1}$. Under the same assumptions as Equation (14) was obtained, we can write for the slopes of the linearized $\sigma_{1}\left(\xi \right)$ and $\sigma_{2}\left(\xi \right)$ functions
(20)$$(\sigma_{Mf1}-\sigma_{Ms1})=\frac{\left(e_{11}-e_{o1}\right)}{\varepsilon^{tr1}}\;\geq \;0$$
and
(21)$$(\sigma_{Mf2}-\sigma_{Ms2})=\frac{\left(e_{12}-e_{o2}\right)}{\varepsilon^{tr2}}\;≅\;0$$
where relations (7) and (8) were also used and, writing Equation (21), we assumed that for a burst-like recovery of M_2_ the slope is approximately zero. Thus, the slope of $\sigma_{1}\left(\xi \right)$ is larger. In addition, consider the case when $\sigma_{Ms1}<$
$\sigma_{Ms2}$ i.e., if
(22)$$\sigma_{o1}\left(T\right)+\frac{e_{01}+d_{o1}}{\varepsilon_{1}}<\sigma_{o2}\left(T\right)+\frac{e_{02}+d_{o2}}{\varepsilon_{2}}$$
which can be fulfilled if *d_o_*_2_ is large enough. Indeed, it can be the case if we assume that the nucleation of M_2_ is more difficult than the nucleation of M_1_. For instance, it is well-known in the literature [33,34] that there can be a competition between habit-plane variants, which can easily nucleate from austenite (and accompanied with smaller transformation strain), and oriented martensites. The nucleation of the latter one is more difficult because of crystallographic compatibility problems, depending on the orientation relationships, and the formation and growth of which can be accompanied with smaller accumulated stresses (see e.g., [9,34,35,36]). The formation of M_2_ at a certain critical stress/strain can happen either from the not transformed yet austenite or from the growing M_1_ (see e.g., [34]).

Figure 3 shows schematically the $\sigma_{1}\left(\xi \right)$ and $\sigma_{2}\left(\xi \right)$ functions (see also Equations (7) and (8)). The intersection of the two straight lines gives the value of *ξ_c_*(~*ε_c_*), at which $\sigma \left(\xi_{c}\right)=\sigma_{Ms2},$ and thus M_2_ can nucleate which leads to a stress drop. For the estimation of the stress drop we can use the results of [22]. It was obtained, for the stress induced austenite to one-martensite-modification transformation, that (due to the transformation strain) there should exists an overall decrease in stress. This decrease, as compared to the pure elastic contribution to the stress–strain curve, is proportional to the product of the martensite volume fraction, ξ, the transformation strain, $\varepsilon^{tr}$, and the effective stiffness in the two-phase region, *S*(*ξ*), as compared to the pure elastic contribution to the stress–strain curve: $\Delta \sigma \;≅-\xi \varepsilon^{tr}S\left(\xi \right)$ (ξ corresponds to the parameter $\xi \;≅\;\frac{n_{z}\left(\varepsilon \right)}{N_{y}}$ used in [22] and $S\left(\xi \right)=\frac{S_{A}S_{M}}{S_{M}+\xi \left(S_{A}-S_{M}\right)}$). Thus, the stress drop due to the nucleation of M_2_ martensite at *ξ_c_* is proportional to $\varepsilon_{2}^{tr}-\varepsilon_{1}^{tr}$:(23)$$\Delta \sigma \;≅-\xi_{c}S\left(\xi_{c}\right)\left(\varepsilon_{2}^{tr}-\varepsilon_{1}^{tr}\right).$$

Now, if the magnitude of Δσ is large enough to decrease the stress below the M_1_ start stress then after the stress drop, the austenite + M_1_ two-phase system will be elastically deformed until the nucleation of M_1_ martensite happens again and the process is repeated. In this (i) case the slope after the stress drop will be the similar as the initial slope of the stress–strain curve, with the effective stiffness *S*(*ξ_c_*). In the opposite case, (ii), after the stress drop there is a further growth of M_1_ (with a similar course of the stress–strain curve as observed during the first stage of the process before the stress-drop) until a new M_2_ nucleates and the process is repeated. Both can be observed in experiments: (i) see Figure 1b of [2] where the segment of the *σ*~*ε* curve after the stress-drop is similar to the initial part, or (ii) in Figure 1a,b in [1] as examples. Figure 1 in [1] also illustrates that with decreasing test temperature there is a transition from (i) to (ii). Regarding the unloading process under compression, with decreasing stress the M_2_ is stable well below the austenite start stress of M_1_, $\sigma_{as1}$, (even below a certain temperature M_2_ remains stable at zero stress). Thus, retwinning of M_2_ (i.e., the formation of M_1_ from M_2_) can start at a certain low stress level and this leads to the expansion of the sample ($\varepsilon_{2}>\varepsilon_{1}$) and a stress jump can appear (see Figure 1b). Further details, like the course of the curve during the stress-drops before reaching the minimum, or where and how M_2_ nucleates can only be explained on the basis of microscopic investigations (see e.g., [35,36,37]) and out of the scope of thermodynamic considerations used here.

Thus, we demonstrated that if the nucleation of the second martensite is difficult, then the interplay of the elastic and dissipative contributions to the *σ*(*ξ*) curve, can be such which can explain the formation of several steps on the *σ*(*ε*) plots. Of course, during the local stress drops the system is also unstable. It can also happen, if e.g., there are only two stress drops, which are frequently observed [2,3], that the presence of these alone can result in an overall negative slope of the stress–strain curves, i.e., the negative slope can be produced by pure nucleation difficulties alone. 

It can be mentioned that experimental results on superelastic behaviour of Fe based alloys sometimes can also show stress drops on the loading branch of the stress–strain curves (see e.g., Figure 6a in [38] or Figure 2d in [39]) and these show quite strong differences in compression and tension. The interpretation of these needs a deeper analysis than the one presented above, where the elastic energy accumulation during martensite formation is handled mainly by its difference when twinned or detwinned martensite modifications grow. In the presence of precipitates and/or retaining martentsites, and when the stress level can be high enough, dislocations can also play a role in the elastic energy accumulation/relaxation as well as in the nucleation barriers and the corresponding microstructure evolution can be quite complex. The better understanding of these phenomena is still in infancy and call for further efforts. 

### 3.4. About the Burst-like Recovery

The shift of the forward transformation peak of the burst-like thermal recovery, as suggested in [12], can happen through re-twinning of M_2_*:* once the twinning is achieved an “easy way” is opened for transformation to the M_1_ or austenite phase(s). Thus, the observed overheating is the consequence of the (nucleation) barrier against re-twinning. We can obtain an approximate expression for the shift of the forward transformation peak of the burst-like recovery as compared to the same thermally induced DSC peak, Δ*T*, using the results described in the preceding sections. Δ*T*, can be given as the difference of the peak temperatures, *T_p_*, during heating:(24)$$\Delta T=T_{p1}-T_{p2}=\frac{A_{f2}+A_{s2}}{-2\Delta s_{2}}-\frac{A_{f1}+A_{s1}}{-2\Delta s_{1}}=\Delta T_{o}+\frac{d_{12}+d_{02}+e_{12}+e_{02}}{-2\Delta s_{2}}-\frac{d_{11}+d_{01}+e_{11}+e_{01}}{-2\Delta s_{1}}$$
where we used that the start and finfish temperatures can be given from (5) at *σ* = 0 (see also [24]. Since the burst-like peak belongs to M_2_*/A* transformation and for this $\left(e_{12}-e_{o2}\right)\;≅\;0$ (see (21)), as well as for two martensite variants we can also assume that the difference of the equilibrium transformation temperatures ($\Delta T_{o}=T_{o2}-T_{o1})$ and the transformation entropy is approximately zero ($\Delta s_{1}\;≅\;\Delta s_{2}=\Delta s$):(25)$$\Delta T\;≅\;\frac{2\left(d_{o2}-d_{01}\right)+2e_{o2}-\left(e_{11}+e_{01}\right)}{-2\Delta s}$$
where it was also used (as in Equations (20) and (21)), that $d_{1i}\;≅\;d_{0i}$ (*i =* 1, 2). In accordance with the previous considerations, we can assume that the difference of the elastic terms is small (and even can be negative, if $e_{11}+e_{o1}>2e_{o2}$ since $e_{11}+e_{o1}>0$) as compared to the first term. Thus, since Δ*T* > 0, the first term dominates in Equation (25). Furthermore, it can also be used that the main difference in the first term is due to the difference of the nucleation energy, i.e., $\left(d_{o2}-d_{01}\right)=\Delta d_{n}>0$, and thus
(26)$$\Delta T\;≅\;\frac{\Delta d_{n}}{-\Delta s}=\frac{D_{2}-D_{1}}{-\Delta s}$$
where the relation $D_{i}=\int_{0}^{1}d_{oi}d\xi =d_{oi}$ (see Equation (13) too) was used and *D_i_* is the nucleation energy for the heating process. Thus, the shift of the transformation peak is a measure of the change in the nucleation energy: $\Delta D=D_{2}-D_{1}.$ As a numerical example we can use the results of [40], where it was obtained that Δ*T* was about 36 K and using the transformation entropy in this alloy, 0.75 J/mol K [40], we find that $D_{2}-D_{1}\;≅\;27\;J/\mathrm{mol}$. This can be compared to the half of the dissipative energy per one thermal cycle, 2.8 J/mol [41], which shows that the nucleation energy for M_2_ can be about an order of magnitude larger than for M_1_.

## 4. Conclusions

The second derivatives of the total Gibbs-free energy, in the framework of a local equilibrium description, were investigated for shape memory alloys showing a burst-like shape recovery after anomalous stress–strain load. We have shown that the thermal hysteresis loops are usually stable and an approximately vertical up branch (i.e., *A_s_* ≅ *A_f_*) can be obtained during burst-like thermal recovery, indicating that the second derivative of the elastic energy is approximately zero in this case. It is also shown that the stress–strain loops for smooth transformations are also stable if only one type of martensite growths. Instability can appear, i.e., the overall slope of the AM branch can be negative, if nucleation and growth of two martensite modifications (variants) takes place and if the second (more stable) one is the final product with $\varepsilon_{2}^{tr}>\varepsilon_{1}^{tr}.$ It is found that local stress-drops, Δ*σ*, on the stress–strain curve can appear if the nucleation of the second martensite is difficult and the presence of few local stress-drops alone can also result in an overall negative slope of the *σ*~*ε* curve. Depending on the magnitude of Δ*σ*, the course of the stress–strain curve after the stress drop is similar to the initial (elastic) part or to the part belonging to *A/*M_1_ transformation. It is illustrated that shift of the temperature of the thermal recovery of M_2_ is a measure of the change in the nucleation energy $\Delta D=D_{2}-D_{1}$.

## Figures and Tables

**Figure 1 materials-15-09010-f001:**
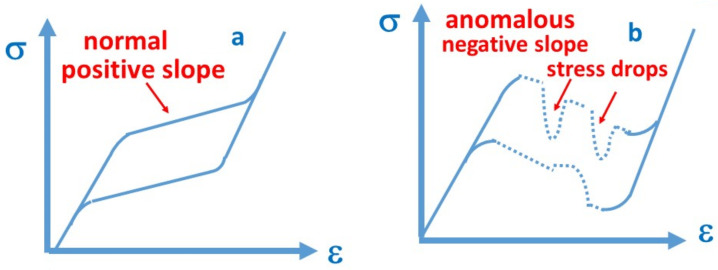
Stress–strain curves schematically: (**a**) typical loop with positive slope of the uploading branch, (**b**) anomalous loop with negative overall slope and stress drops on it.

**Figure 2 materials-15-09010-f002:**
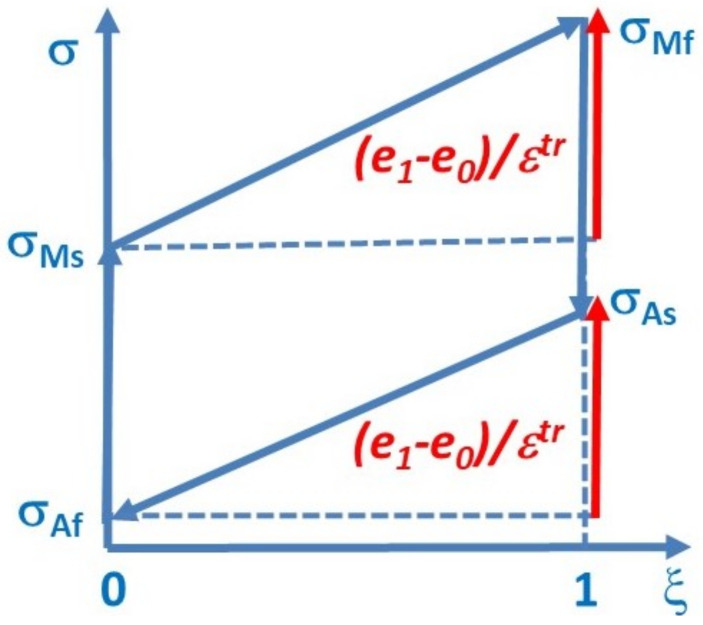
Schematic *σ*(*ξ*) hysteresis loop for *ε^tr^* = *const.* > 0. Since *ε* = *ξε^tr^*, with the actual value of *ε*, this plot corresponds to the *σ* versus *ε* plots.

**Figure 3 materials-15-09010-f003:**
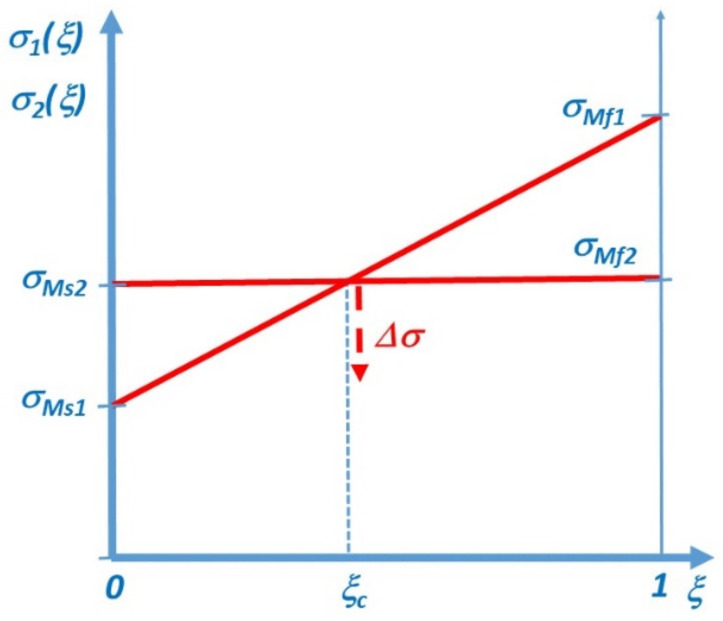
Schematic $\sigma_{1}\left(\xi \right)$ and $\sigma_{1}\left(\xi \right)$ functions, and the stress drop, Δσ at the critical volume fraction where the second martensite nucleates (see also the text).

## Data Availability

Not applicable.

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
