# Peer review of "Thermodynamic Analysis of Anomalous Shape of Stress–Strain Curves for Shape Memory Alloys"

_materials, 2022, doi:10.3390/ma15249010_

Round 1

Reviewer 1 Report

In this paper the authors investigated the negative slope of stress-strain curve (and stress drops) of stress-induced Austenite - Martensite transformation in shape memory alloys. They proposed a mechanism for such observation – the growth of a second martensitic variant may cause such stress-strain observed behaviors. This study is interesting and inspiring. The following raised issues may help the authors clarify certain issues and improve the manuscript:

1. Please discuss the possibility of the observed stress drop due to the noise coming from the operation in the test.

2. The authors seem to simplify the problem as a single-crystal single-domain problem. What’s the space distribution for the martensitic variant? The authors proposed a two-martensite scenario. What happens if we look at a more realistic case which includes various types of Martensitic variants? The authors should provide a complete stress-strain loop generated according to their proposed model. Right now I only see schematics, which is not sufficient to support their conclusions.

3.  Eq. (4) is a key equation that reflects the hypothesis made in this study. This relation is quite similar to other phenomenological approaches, such as phase field method (Xi and Su, 2021. Acta Mechanica, 232(11), 4545-4566), wherein the parameter η is equivalent to an order parameter. However, in phase-field approach, multiple martensitic variants can be simultaneously modeled, and the order parameters are not associated with the total volume fraction of the Martensite. The authors should discuss the difference between these two approaches or provide some insights.  

Reviewer 2 Report

This paper uses the thermodynamic analysis of superelastic stress-strain curve. The theoretical frameworks are well presented. Minor issues need to revise in the paper.

1.      The author concludes the stress-strain loops for smooth transformations are stable if only one type of martensite growths and the slope can be negative. If two martensite variants take place, the local stress-drops can appear if the nucleation of the second martensite is difficult.

In FeNiCoAlTa and FeMnAlNi single crystal studies, the smooth stress-strain curve is observed when two martensite can assist the deformation. On the other hand, only one martensite variant can assist the deformation, the stress-strain curve shows the stress-drop and large irrecoverable strain. The experiment results of iron-based shape memory alloys seem different from the theoretical analysis of present paper.

Could the author explain the reason? Or this theoretical formula can only use in NiFeGaCo system

2.      Can the theoretical model apply in different martensite transformation system?

3.      Can this thermodynamic analysis in polycrystalline or only work in single crystal with <100> orientation?

4.      The conclusion title changes the number to 4.

Round 2

Reviewer 1 Report

The paper is now considered publishable